

**Effects of tropical rainforest conversion to rubber plantation**
**on soil quality in Hainan Island, China**
Rui Sun[a,b], Guoyu Lan[a,b*], Chuan Yang[a,b], Zhixiang Wu[a,b], Banqian Chen[a,b], Klaus
Fraedrich[c]
[a] Rubber Research Institute, Chinese Academy of Tropical Agricultural Sciences,
Haikou 571101, China
[b] Hainan Danzhou Agro-ecosystem National Observation and Research Station,
Danzhou 571737, China
[c] Max Planck Institute for Meteorology, Hamburg 20146, Germany
[*] **Corresponding Author:**
Guoyu Lan, No. 4, Xueyuan Road, Longhua District, Haikou 571101, China. Email:
langyrri@163.com.





**Abstract**
Land-use changes can alter soil properties and thus affect soil quality. Our
understanding of how forest conversion (from tropical rainforest to rubber plantations)
affects soil properties and soil quality is limited. An ideal testing ground for analyzing
such land-use change and its impacts is Hainan Island, the largest tropical island in
China. Based on 21 soil physicochemical and biological properties, a soil quality
index (SQI) employed principal component analysis to assess soil quality changes
from the conversion of tropical rainforests to rubber plantations. The results showed
that (i) soil available potassium, available phosphorus, microbial biomass carbon,
cellulose decomposition, acid phosphatase, and urease were vital soil properties for
soil quality assessment on Hainan Island. (ii) The SQI of rubber plantations decreased
by 26.48% compared to tropical rainforests, while four investigated soil properties
(soil pH, total phosphorus, cellulose decomposition, and actinomyces) increased. (iii)
The SQI of both the tropical rainforests and rubber plantations showed significant
spatial differences, which, under tropical rainforests, was more sensitive to seasonal
changes than those under rubber plantations. (iv) Structural equation modeling
suggested that forest conversion directly impacted soil quality and, indirectly
impacted soil qualities' spatial variation by their interaction with soil types and
geographical positions. Overall, though the conversion of tropical rainforest to rubber
plantation did not decrease all soil properties, the tropical rainforest with its high soil
quality should be protected.
**Keywords:** rubber plantation; tropical rainforest; soil properties; soil quality index;





structural equation modeling

## 1. Introduction

The rubber tree (*Hevea brasiliensis*), an economically valuable forest species, is
a large source of natural rubber and is grown in more than 40 tropical countries
worldwide (Warren-Thomas, 2015). Due to the increasing development of tire
manufacturing and high prices of rubber, the land-mass of rubber plantations has
expanded rapidly over the last 20 years in tropical Asia (Ahrends et al., 2015;
Warren-Thomas et al., 2015; Lang et al., 2017), which is currently the world's most
prolific region for rubber production (FAO, 2017). The conversion of tropical
rainforests to rubber plantations in tropical Asia accompanies the continuously rising
demand for rubber worldwide (De Blécourt et al., 2014; Allen et al., 2015; Hassler et
al., 2017; Guillaume et al., 2018), which generally has negative impacts on soils and
ecosystem services and threatens biodiversity and human livelihoods (Qiu, 2009;
Ziegler et al., 2009; Tan et al., 2011; Ahrends et al., 2015; Liu et al., 2019; Singh et al.,
2021). Hence, the response of rubber plantation expansion at the expense of tropical
rainforest degradation on the environment - especially soil quality - has recently
become a research focus.
Soil quality is a critically important capacity of soil within ecosystems,
functioning not only as sustenance for biological productivity but also in the
maintenance of environmental quality and promotion of plant and animal health
(Doran and Parkin, 1994; Karlen et al., 2003; Shao et al. 2020; Li et al. 2020). Soil
quality can be assessed based on a set of soil properties that affect soil functions (Lal,





1998), such as physical, chemical, and biological soil properties (Yakovchenko et al.,
1996; Gil-Sotres et al., 2005; Griffiths et al., 2010; Nosrati et al., 2011; Davari et al.,
2020). Evaluating soil quality generally involves three main steps: definition and
selection of soil properties, scoring soil properties, and soil quality index calculation
(Andrews et al., 2004; Chen et al., 2013). At present, the comprehensive soil quality
evaluation methods mainly include soil quality cards and test kits (Ditzler and Tugel,
2002), grey correlation method (Yang et al., 2010), fuzzy methods (Torbert et al.,
2008; Yue-Ju et al., 2010; Xue et al., 2010), soil quality indices (Andrews et al., 2004;
Masto et al., 2008; Nakajima et al., 2015; Nabiollahi et al., 2017; Zhang et al., 2019;
Shao et al., 2020; Jahany and Rezapour, 2020; Jin et al., 2021), and soil quality index
area (Kuzyakov et al., 2020). Out of all the methods, the soil quality index (SQI)
approach has been applied frequently because of its flexibility and simplicity
(Derakhshan-Babaei et al., 2021;  Zhang et al., 2021).
Soil quality can be affected by land use and land-use changes (Marzaioli et al.,
2010; Moges et al., 2013; Raiesi, 2017; Yu et al., 2018; Pham et al., 2018; Davari et
al., 2020). Particularly, land-use changes can influence soil degradation (Vityakon,
2007; Nabiollahi et al., 2018; Nosrati and Collins, 2019), soil physical and chemical
quality (Deng et al., 2016; Liu et al., 2018; Wang et al., 2019; Sun et al., 2021), soil
biological quality (Berkelmann et al., 2018; Cai et al., 2018), etc. The conversion of
tropical rainforests to rubber plantations, a typical land-use change in tropical regions,
has been the focus of many previous studies looking at the land-use change affecting
soil properties and functions, such as soil physical and chemical properties (Chen et



al., 2019; Sun et al., 2021), soil nutrients and fertility (Chiti et al., 2014; De Blécourt
et al., 2014; Guillaume et al., 2015; Allen et al., 2015; Hassler et al., 2017; Maranguit
et al., 2017), soil respiration (Goldberg et al., 2017; Zhao et al., 2018), and soil
microbial communities (Krashevska et al., 2015; 2019; Kerfahi et al., 2016; Wang et
al., 2017; Berkelmann et al., 2018; Lan et al., 2017a; 2020). A soil quality index was
also established based on soil's physical and chemical properties to comprehensively
assess the effects of rubber plantations on the soil after being converted from tropical
rainforest (Sun et al., 2021; Zou et al., 2021). Chemical properties were found to
contribute more to the soil quality index than the physical properties (Sun et al., 2021;
Zou et al., 2021), and biological properties were rarely considered.
Hainan Island is the largest tropical island in China. It is a major producer of
natural rubber, with an output of 350.68 million kg from an area of 5,283.51 km$^2$
under rubber cultivation in 2018 (Statistical Bureau of Hainan Province, 2019).
During the past few decades, the size and number of rubber plantations have been
expanding rapidly on the island at the expense of losing forested land and agricultural
land (Zhai et al., 2012; 2014; Chen et al., 2016; Sun et al., 2020), which overall has
decreased tropical rainforest area. Therefore, Hainan Island was recognized as an
ideal testbed for analyzing this specific land-use change (from tropical rainforests to
rubber plantations) and its impacts. A soil quality index (SQI) based on a weighted
summation of soil physical, chemical, and biological properties was established in this
study, aiming (i) to assess soil quality of tropical rainforests and rubber plantations
comprehensively, and (ii) to quantify the impact of the land-use change (from tropical



rainforests to rubber plantations) on soil quality variations on Hainan Island. Thereby,
tests of the following hypotheses are required: (i) soil quality, which deteriorates by
the conversion of tropical forests to rubber plantations, and (ii) the spatial variation of
soil quality, which is affected by the interaction of land-use change with soil types,
geographical position, and climatic variables.
**2. Materials and methods**
**2.1. Study area**

Hainan Island (18°09′-20°10′ N and 108°37′-111°03′ E, Fig. 1) is the largest

island in Southern China, with a geographical area of 33,920 km$^2$. It is also the largest
island in the Indo-Burmese biodiversity hotspot (Myers et al., 2000; Wikramanayake
et al., 2002), characterized by a tropical monsoon climate. The climate is warm and
humid, with a rainy season from May to October and a dry season from November to
April (Wu, 2008; Sun et al., 2017). The annual average temperature varies from 23.4
~ 24.7 °C across the study area, while the mean annual precipitation ranges from
1392.3 mm to 2173.8 mm.

The central part of Hainan Island is mountainous, containing primary forest

composed of a mix of tropical rainforest and monsoon forest. The tropical forest,
accounting for 17.3% of the island's area, is mainly distributed in the mountains in the
south-central region at altitudes above 500 m. Rubber plantations are located in the
lowlands surrounding the central mountainous area, where transportation is more
accessible, and water sources are nearby (Sun et al., 2020; Fig.1).

The study sites included four different soil types: laterite, lateritic red soil, red




soil, and yellow soil. The soil data (with a resolution of 1:1 000 000), was obtained
from a soil survey (completed in 1995 by the National Soil Survey Office of China)
and from the Resources and Environment Data Cloud Platform
(http://www.resdc.cn/data.aspx?DATAID=145).
**2.2. Soil sampling and experimental design**

Soil samples were collected from tropical rainforest and rubber plantation on

Hainan Island in January 2018 and July 2018. Five sites were selected for this study
representative of the major tropical rainforest districts of Hainan (Fig. 1), i.e., Bawang
mountain (BW), Diaoluo mountain (DL), Wuzhi mountain (WZ), Yinge mountain
(YG), and Jianfeng mountain (JF). For rubber plantations, five sites were also selected
in the northeast, northwest, center, southeast and southwest of the island, respectively:
Haikou (HK), Danzhou (DZ), Qiongzhong (QZ), Wanning (WN), and Ledong (LD).
Note that, mature rubber plantations (25 to 30 years of age) were chosen for each site
to avoid the variable rubber plantation age on soil quality. And intensive management
practices were utilized in rubber plantations, such as latex harvest, and the application
of fertilizers (Lan et al., 2017). In order to facilitate latex harvest, rubber trees were
fertilized once or twice a year using compound fertilizer at a rate of 1-1.5 kg per tree
and organic fertilizers at a rate of 20-25 kg per tree.

Study sites characteristics are given in Table 1. For each site, thirteen sample

plots were selected within an area of one square kilometer. A five-point sampling
method was used, and compound soil samples were obtained from each plot. There
were a total of 65 samples collected from both the rubber plantation and tropical



rainforest sites. We sampled twice, once in the rainy season (July) and once in the dry
season (January), making for a total of 260 soil samples (130 from rubber plantations
and 130 from the tropical rainforest). After removing the litter layer, using a sterilized
steel drill, a 5-cm diameter soil core was collected from 0 to 20 cm depth,
homogenized, and passed through a 2-mm mesh sieve. Soil samples for
physicochemical analysis were put in a sterilized self-sealing bag and stored at 4 °C
and were then transported to the laboratory for analysis.
**2.3. Soil analysis**
A total of twenty-one soil physical (WC: soil water content), chemical (pH, SOM:
soil organic matter, TN: total nitrogen, NN: nitrate-nitrogen, AN: ammonium nitrogen,
TP: total phosphorus, AP: available phosphorus, TK: total potassium, and AK:
available potassium), and biological properties (MBC: microbial biomass carbon, RQ:
microbial respiratory quotient, CD: cellulose decomposition, URE: urease, ACP: acid
phosphatase, CAT: catalase, CEL: cellulose, SI: sucrose invertase, BAC: bacteria,
FUN: fungi, and ACT: actinomyces) were determined to be soil indicators for soil
quality assessment.
Soil physical and chemical properties were quantified using standard techniques
recommended by a guide to soil physical and chemical analysis (Institute of Soil
Science, Chinese Academy of Sciences, 1978). Detailed protocols for measuring soil
water content are available (see Deng et al. 2016; Chen et al., 2019; Zhang et al.,
2019). Soil pH was measured in a 1:1 soil-water suspension with a pH meter (pHS-2,
Leici, China). Soil organic matter was determined by the potassium dichromate



oxidation method. TN was determined using micro-Kjeldahl digestion followed by
steam distillation. NN and AN were determined by steam distillation and
indophenol-blue colorimetry, respectively. TP and AP were quantified using the
molybdenum-antimony anti-spectrophotometric method; TK and AK were measured
by flame photometry (Soil Science Society of China, 2000).

Soil biological properties were measured, including soil microbial function

(MBC, RQ, and CD), soil microbial quantity (BAC, FUN, and ACT), and enzymatic
activity (URE, ACP, CAT, SI, and CEL). MBC was measured by the chloroform
fumigation method (Ross, 1990). RQ was titrated by alkali absorption, and CD was
decomposed by the embedding sheet method (Xu and Zheng, 1986). URE, ACP, CAT,
CEL, and SI were determined using sodium phenol sodium hypochlorite colorimetry,
colorimetric method of benzene disodium phosphate, potassium permanganate
titration, nitrosalicylic acid colorimetry, and the 3, 5-dinitrosalicylate colorimetric
method, respectively (Guan, 1986). Measurement of bacteria, fungi, and actinomyces
included DNA extraction and PCR amplification using Illumina MiSeq sequencing
and bioinformatic analysis pipelines referred to in a previous study (Lan et al., 2020).
**2.4. Soil quality assessment method**

Based on the twenty-one physical, chemical, and biological soil properties, a soil

quality index (SQI) was established employing principal component analysis (PCA)
to comprehensively assess soil qualities in spatial variation and seasonal changes
under tropical rainforests and rubber plantations on Hainan Island.

First, all the selected soil properties were scored using the scoring function





"more is better" (Andrews et al., 2002; Shao et al., 2020) according to the soil
functions of each soil property. The equation of the scoring function "more is better"
(Eq. (1)) is as follows:


$$f(x) = \begin{cases} 0.1, & x \leq L \\ 0.9 \times \dfrac{x-L}{U-L} + 0.1, & L < x < U \\ 1, & x \geq U \end{cases} \qquad (1)$$

where $f(x)$ is the linear score of soil properties, $x$ is the value of soil properties, and $L$
and $U$ are the lower and upper threshold values of the property, respectively.

Second, all the soil properties of tropical rainforests and rubber plantations were

grouped into components for PCA. The weights of the properties were calculated
using Eq. (2) based on the values of their communalities. Communality describes the
proportion of variance in each soil property explained by the PCA model. The larger
the communality, the higher the proportion of an indicator's variance can be explained
by the factors (Brejdaet al., 2000; Imaz et al., 2010; Chen et al., 2013; Zhang et al.,

2016).

$$W_i = C_i \Big/ \sum_{i=1}^{n} C_i \qquad (2)$$

where $W_i$ is the weight of the soil properties, $C_i$ is the communality value of soil
property obtained from the PCA results, and $n$ is the number of soil properties.

Finally, after all the soil properties were scored and weighted, SQIs were

calculated using the PCA-based Soil Quality Index equation (Eq. (3)):



$$SQI = \sum_{i=1}^{n} W_i P_i \qquad (3)$$


where $W_i$ is the weight of each soil property, $P_i$ is the score of each soil property, and
$n$ is the number of soil properties.

Except for calculating the weight of each soil property, the PCA can be used to

select vital soil properties for assessing soil quality (Ngo-Mbogba et al., 2015).
Principal components (PCs) are sets of indicators with large eigenvalues and factor
loading. Only the PCs with eigenvalues $\geq 1$ (Brejda et al., 2000) and PCs that
explained at least 5% of the variation in the data (Mandal et al., 2008) were selected.
According to Andrews and Carroll (2001), soil properties with weighted absolute
values within 10% of each PC's the highest soil property value were selected.
However, in the process of calculating the factor load of each soil property by PCA
and the data structure is simplified, some important soil properties' information can be
lost (Yemefack et al., 2006). The norm value, which is the magnitude (length) of the
vector representing the variable in the multi-dimensional space spanned by the set of
PCs, was introduced to avoid this defect (Yemefack et al., 2006). The higher the norm
value is, the stronger its ability to represent the overall soil quality information for
further interpretation. The equation of the norm is:

$$N_{ik} = \sqrt{\sum_{i}^{k} \left( U_{ik}^2 \lambda_{ik} \right)} \qquad (4)$$


where $N_{ik}$ is the comprehensive loading of the i-th soil variable on the first $k$ PCs,
$\lambda_{ik}$ is the eigenvalue of the PC, and $U_{ik}$ is the loading of the i-th soil variable on
$PC_k$. Soil properties receiving $N_{ik}$ within 10% of the highest norm values were





considered the most important for assessing soil quality (Chen et al., 2013; Zhang et
al., 2016).

**2.5. Statistical analysis**

One-way analyses of variance (ANOVA) and *Tukey* HSD post hoc tests were

used to assess the significant difference (P < 0.05) of the investigated soil properties
and soil quality between tropical rainforests and rubber plantations in wet and in dry
seasons. A radar diagram was drawn to show each of the soil properties, and soil
functions changed by converting tropical rainforests to rubber plantations (Kuzyakov
et al., 2020). Pearson correlation analysis (PeCA) was conducted to identify
relationships among measured soil properties. Structural equation model (SEM) was
established to reveal hypothetical relationships based on the assumption that the
spatial variation of soil quality is affected by land-use change with soil types,
geographical position, and climatic variables.

**3. Results**

**3.1. Soil properties under tropical rainforests and rubber plantations**

Descriptive statistics of the measured soil properties are shown in Table 2. Soil

pH was acidic in the investigated rubber plantations and tropical rainforests. Most soil
chemical and biological properties of tropical rainforests were significantly higher
than those of rubber plantations, such as SOM, TN, TK, NN, AN, AP, AK, RQ, URE,
ACP, CAT, SI, and CEL. However, Soil pH, TP, CD, and ACT, increased noticeably
with the conversion of natural tropical rainforest to monoculture rubber plantations.

Soil TN, TP, AP, URE, and SI varied significantly between seasons in both





rainforests and rubber plantations; the concentration of these properties increased
from dry to rainy seasons. These results suggested that seasonal patterns substantially
affected the chemical and biological properties of the soil, and thus, the soil quality.
**3.2. Soil quality changes from tropical rainforests to rubber plantations**
SQI values of the investigated tropical rainforest and rubber plantations on Hainan
Island were calculated based on the soil property score and weights (Table 3), ranging
from 0.358 to 0.418 for tropical rainforests and from 0.229 to 0.325 for rubber
plantations (Fig. 2a). The SQI values of tropical rainforests were significantly higher
than rubber plantations (P < 0.05), which indicated that the conversion of natural
tropical rainforest to monoculture rubber plantations would deteriorate soil quality.
For the seasonal difference, the wet season SQI values were significantly higher than
those in the dry season under tropical rainforests conditions. At the same time, there
were no significant differences for rubber plantations (Fig. 2b), indicating that the soil
quality under tropical rainforests was more sensitive to seasonal changes than those
under rubber plantations.

To show each soil property and soil functions change by the conversion from

tropical rainforests to rubber plantations, a radar diagram was constructed for both the
soils under tropical rainforests and rubber plantations (Fig.3), assuming the averaged
soils under tropical rainforests as natural soil. Our study found that most soil
properties and functions decreased when converting tropical rainforests to rubber
plantations. In contrast, soil TP, CD, and ACT increased by 59%, 91%, and 94%,
respectively (Fig.3). In addition, the radar diagram indirectly reflects the most



sensitive (AK, ACP, CEL, TP, CD, ACT) and resistant (WC, MBC) soil properties by
comparing the soils of rubber plantations and tropical rainforests (Fig. 3).
**3.3. Important soil properties for soil quality assessment**
Important soil properties for SQI values under the tropical rainforests and rubber
plantations on Hainan Island were determined based on the absolute factor loading
values ($\geq$0.50) of each PC and the norm values (within 10% of the highest values)
(Shao et al. 2020). PeCA examined the relationships among these properties to reduce
redundancy (Table 4). PCA results showed that the first six components had
eigenvalues>1, with values ranging from 1.168 to 5.771, each explaining at least
5.561% of the data variation and accounting for 68.539% of the total variance (Table
3). Thus, the first six components were selected. In PC1, the absolute factor loading
values of NN, AK, SOM, pH, TN, URE, ACP, CAT, SI, and CEL were $\geq$0.50. Among
these soil properties, ACP had the highest norm value of 2.08, NN, SOM, TN, and
CEL had norm values within 10% of the highest value. As ACP, NN, SOM, TN, and
CEL significantly correlated, ACP was selected as the first important soil property.
Similarly, the other five components, AK, AP, MBC, CD, URE, and FUN, were also
selected.
The accuracy analysis of the selected soil properties for quality assessment,
SQI-M (including ACP, AK, AP, MBC, CD, and URE), showed that the SQI-M values
significantly correlated with SQI values of the total soil properties (Fig. 4a). From the
six soil properties, ACP contributed 26.91% to SQI-M, followed by URE (15.82%),
AK (14.18%), MBC (13.58%), and AP (10.01%), FUN and TP had the lowest

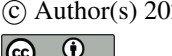



contribution (9.79% and 9.71%) (Fig.4b).
**3.4. Factors influencing soil quality**
A structural equation model was established to explain the relationships between
the soil quality index and its influential factors (Fig. 5). The influential factors, which
may drive variation in soil quality, are related to climate (temperature and
precipitation), geographical location (latitude, longitude, and altitude), land-use
change, and soil type. Our structural equation model explained 57% of the variation in
SQI values for the tropical rainforests and rubber plantations on Hainan Island. The
land-use change (from rubber plantations to tropical rainforests) played the most
significant positive role in the spatial variation of SQI, followed by the climate.
Land-use type, soil type, and geographical position also interacted with each other.
Hence, there were some direct and indirect effects of land-use type on the soil quality.
**4. Discussion**
**4.1. Soil properties affected by tropical rainforests converted to rubber**
**plantations**
The conversion of tropical rainforests to rubber plantations decreased most soil
chemical and biological properties on Hainan Island. Soil nutrient status (SOM, TN,
TK, NN, AN, AP, and AK), soil microbial function (RQ), soil microbial quantity
(BAC and FUN), and enzyme activities (URE, ACP, CAT, SI, and CEL), generally
displayed a net lower level in rubber plantations than in tropical rainforests, which
was consistent with many previous studies (Allen et al., 2015; Balasubramanian et al.,
2020; Singh et al., 2021).





However, four investigated soil properties, i.e, soil pH, TP, CD, and ACT, were
demonstrated to increase by converting tropical rainforest to rubber plantations on
Hainan Island. The soil pH change of tropical forests to rubber plantations was
consistent with the previous studies in Sumatra, Indonesia (Allen et al., 2015; 2016),
opposite the earlier study of the Xishuangbanna region of Yunnan Province, China
(Liu et al., 2019). The soil TP in rubber plantations was greater than the adjacent
native forest on Hainan Island. The greater TP concentrations on rubber plantations
could be caused by fertilization and the net transfer of phosphorus from dead
vegetables, litter, and decaying roots to soil (Yang et al., 2010a; Wang et al., 2017).
**4.2. Soil quality affected by tropical rainforests converted to rubber plantations**
Our previous study has found that the comprehensive assessment indices based
on fourteen soil physical and chemical properties of rubber plantations were
significantly lower than those of tropical rainforests on Hainan Island (Sun et al.,
2021). Similarly, as in Xishuangbanna (southwest China) the soil quality index value
based on 23 soil physical and chemical properties of the rubber plantation decreased
by 15.50%, compared to the primary rainforest (Zou et al., 2021). The previous
studies also found that chemical properties contributed more to the soil quality index
than the physical properties (Sun et al., 2021; Zou et al., 2021), with the biological
properties were rarely considered. Hence, soil chemical and biological properties were
the focus of this study. And the results indicated that the soil quality index value of the
investigated rubber plantations decreased by 26.48%, compared to the primary
rainforests on the tropical island.



Comparing various SQI between studies is complex and partly impossible
because of the diverse soil properties and weighting factors (Kuzyakov et al., 2020).
Hence, a radar diagram was applied to show each of the soil properties and soil
function changes. It was found that most of the soil properties and functions decreased
by the conversion of tropical rainforests to rubber plantations on Hainan Island.
Taking into account that the soil quality would significantly decrease from high to low
plant diversity on rubber plantations (Hemati et al., 2020), interplanting (Liu et al.,
2018a, 2019; Chen et al., 2019; Sun et al., 2021; Zou et al., 2021) and natural
management (Lan et al., 2017a) were considered as alternative mechanisms to
improve soil quality on monoculture rubber plantations.
**4.3. Factors influencing soil quality**
Land-use change (from rubber plantations to tropical rainforests), interacting with soil
type and geographical position, played the most critical positive role on the SQI
variation (Fig.5). SQI variation illustrated that soil quality was negatively affected by
the conversion of tropical rainforests to rubber plantations. The spatial variation of
SQI was significant on both the rubber plantations and tropical rainforests of Hainan
Island (Fig. 2a), indicating that spatial variability played an important role in soil
quality. Previous studies have been found that spatial variability (e.g., soil depth
intervals) surpasses land-use change effects on soil biochemical properties of
converted lowland landscape in Sumatra, Indonesia (Allen et al., 2016).
Seasonal changes also played a role in soil quality. According to the SQI values,
tropical rainforests in the wet season were significantly higher than those in the dry



season. However, there were no significant differences in SQI values for rubber
plantations, which can be attributed to the fertilization in dry seasons. Although the
effect of seasonal change on the SQI values under rubber plantations was relatively
small, it controlled some important soil chemical and biological properties (e.g., TN,
TP, AP, URE, and SI) as well as the bacterial communities in soils of rubber
plantations in tropical region of Hainan (Lan et al., 2018; 2020).
**5. Conclusions**
The soil quality of rubber plantations decreased compared to the tropical rainforests
on Hainan Island, with soil AK, AP, MBC, CD, ACP, and URE as vital soil properties.
However, four investigated soil chemical and biological properties (soil pH, TP, CD,
and ACT) increased by the conversion of tropical rainforest to rubber plantations.
Except for the land-use change, spatial variability and seasonal changes played
essential roles in soil quality, and soil quality under tropical rainforests was more
sensitive to seasonal changes than rubber plantations. In this sense, the conversion of
tropical rainforest to rubber plantations results in significant changes in soil quality;
thus, the tropical rainforest with its high soil quality should be protected.
**Declaration of Competing Interest**
The authors declare that they have no known competing financial interests or personal
relationships that could have influenced the work reported in this paper.
**Acknowledgments**
We would like to acknowledge the financial support of the High-level Talent Project



of Hainan Basic and Applied Basic Research Program (Natural Science) (2019RC335;
320RC733), National Natural Science Foundation of China (31770661), Finance
Science and Technology Project of Hainan Province (ZDYF2019145), and the
Earmarked Fund for China Agriculture Research System (CARS-33-ZP3).

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





**List of Figures**
**Fig. 1** Maps of the geographic position, topography, and soil sampling sites of Hainan
Island, China: (a) location of Hainan Island (red); (b) topography and drainage of
Hainan Island; (c) spatial distribution of soil sampling sites in tropical rainforests and
rubber plantations.
**Fig. 2** Soil quality index (SQI) values under rubber plantations and tropical rainforests
on Hainan Island: (a) spatial distribution; (b) temporal variation. Different lower-case
(or upper-case) letters indicate significant difference at $P < 0.05$ between the seasonal
(or annual) SQI values of rubber plantations and tropical rainforests.
**Fig.3** Radar diagram for soil properties changing by the conversion from tropical
rainforests to rubber plantations on Hainan Island. The measured soil properties are:
WC, soil water content; SOM, soil organic matter; PH, soil pH; TN, total nitrogen;
NN, nitrate nitrogen; AN, ammonium nitrogen; TP, total phosphorus; AP, available
phosphorus; TK, total potassium; AK, available potassium; MBC, microbial biomass
carbon; RQ, microbial respiratory quotient; CD, cellulose decomposition; NF,
nitrogen fixation; UR, urease; ACP, acid phosphatase; CAT, catalase; CEL, cellulose ;
SI, sucrose invertase; BAC, bacteria; FUN, fungi; ACT, actinomyces.
**Fig.4** (a) Scatter diagram and linear relationships between SQI-M and SQI values (n =
260) and (b) individual contributions of soil properties to the soil quality indicator
SQI-M based on the seven important properties (SQI is the soil quality indicator
based on the total soil properties). The measured soil properties are: AP, available
phosphorus; AK, available potassium; MBC, microbial biomass carbon; CD, cellulose





decomposition; UR, urease; ACP, acid phosphatase; FUN, fungi.
**Fig.5** Structural equation model (SEM) analysis of the effects of land-use changes,
soil types, climatic variables, and geographic position on the soil quality index (SQI).
Red arrows indicate negative effects and green arrows represent positive effects.
Numbers adjacent to arrows are path coefficients ($p$ values) indicating the effect size
of the relationship, and $p$ values are as follows: *$p$ <0.05; **$p$ <0.01; ***$p$ <0.001.
CFI: Comparative Fit Index; RMSEA: Root Mean Square Error of Approximation.









**Table 1** Site characteristics for tropical rainforests and rubber plantations.

| Site name | Longitude (°) | Latitude (°) | Elevation (m) | Forest type | Soil type | Precipitation (mm) | Temperature (℃) |
|---|---|---|---|---|---|---|---|
| Danzhou (DZ) | 109.58 | 19.56 | 112 | Rubber plantation | Laterite | 1831.5 | 23.6 |
| Qiongzhong (QZ) | 109.74 | 19.26 | 156 | Rubber plantation | Lateritic red soil | 2067.3 | 23.5 |
| Ledong (LD) | 109.22 | 18.75 | 170 | Rubber plantation | Laterite | 1661.3 | 24.5 |
| Wanning (WN) | 110.13 | 18.67 | 51 | Rubber plantation | Laterite | 1786.5 | 24.7 |
| Haikou (HK) | 110.57 | 19.70 | 102 | Rubber plantation | Laterite | 1863.4 | 24.2 |
| Diaoluo (DL) | 109.86 | 18.73 | 958 | Rainforest | Red soil | 1921.3 | 24.2 |
| Jianfeng (JF) | 108.88 | 18.73 | 950 | Rainforest | Yellow soil | 1392.3 | 24.7 |
| Bawang (BW) | 109.13 | 19.08 | 575 | Rainforest | Red soil | 1602.1 | 24.3 |
| Yingge (YG) | 109.56 | 19.05 | 620 | Rainforest | Red soil | 2067.8 | 23.6 |
| Wuzhi (WZ) | 109.68 | 18.91 | 820 | Rainforest | Yellow soil | 2173.8 | 23.4 |

Notes: Soil type data was obtained from a soil survey (completed in 1995 by the National Soil Survey Office of China) and from the
Resources and Environment Data Cloud Platform (http://www.resdc.cn/data.aspx?DATAID=145). The precipitation and temperature data
of each site was obtained from the results of a reference (Sun et al. 2016).





Table 2 Soil properties under tropical rainforests and rubber plantations on Hainan
Island.

| Soil properties | | Rubber plantation | | | Tropical rainforest | | |
|---|---|---|---|---|---|---|---|
| | | Dry season | Wet season | Annual | Dry season | Wet season | Annual |
| WC | % | 32.08±9.23a | 28.17±9.65a | 30.13±9.61A | 31.08±11.13a | 32.17±8.29a | 31.63±9.79A |
| AN | mg/kg | 12.28±5.03a | 12.49±6.92a | 12.39±6.03A | 14.18±4.03ab | 16.45±6.46b | 15.31±5.49B |
| NN | mg/kg | 7.35±3.79a | 6.53±3.17a | 6.94±3.50A | 10.34±3.93b | 15.69±7.86c | 13.01±6.75B |
| AP | mg/kg | 2.62±0.95a | 3.92±2.61b | 3.27±2.06A | 1.79±0.48ab | 7.1±6.13c | 4.45±5.09B |
| AK | mg/kg | 27.12±13.05a | 31.63±17.15a | 29.37±15.35A | 72.23±36.82b | 84.92±41.64b | 78.58±39.67B |
| SOM | % | 1.48±0.86a | 1.65±0.89a | 1.56±0.88A | 2.68±0.95b | 3.06±0.67b | 2.87±0.84B |
| pH | | 4.71±0.34ab | 4.87±0.51b | 4.79±0.44A | 4.6±0.33a | 4.59±0.76a | 4.59±0.59B |
| TN | g/kg | 0.94±0.39a | 1.62±0.69c | 1.28±0.65A | 1.34±0.34b | 2.82±0.74d | 2.07±0.94B |
| TP | (P$_2$O$_5$)% | 0.06±0.04b | 0.07±0.04c | 0.06±0.04A | 0.03±0.01a | 0.05±0.04b | 0.04±0.03B |
| TK | (K$_2$O)% | 1.18±1.09a | 1.47±1.15ab | 1.33±1.12A | 1.73±0.74b | 1.41±0.79ab | 1.57±0.78B |
| MBC | mg/kg | 0.07±0.02b | 0.04±0.05a | 0.05±0.04A | 0.04±0.04a | 0.05±0.06ab | 0.05±0.05A |
| RQ | mg/kg | 115.91±67.15a | 118.31±83.11a | 117.11±75.27A | 218.98±75.95b | 117.28±53.48a | 168.13±82.99B |
| CD | % | 0.66±0.38bc | 0.68±0.43c | 0.67±0.41A | 0.51±0.36b | 0.19±0.09a | 0.35±0.31B |
| UR | mg/kg | 46.28±32.52a | 71.95±37.19b | 59.11±37.11A | 62.16±25.66ab | 109.7±52.83c | 85.93±47.46B |
| ACP | mg/kg | 1522.2±543.98b | 1007.04±513.99a | 1264.62±587.14A | 3424.72±464.48c | 3264.83±982.59c | 3344.77±769.72B |
| CAT | ml/g | 0.69±0.31b | 0.46±0.29a | 0.57±0.32A | 1.09±0.45c | 0.94±0.26c | 1.01±0.37B |
| SI | mg/kg | 2714.24±1648.49a | 4398.23±1768.33b | 3556.23±1901.06A | 5956.87±2971.49c | 8156.99±4850.99d | 7056.93±4156.32B |
| CEL | mg/kg | 20.15±19.15a | 26.05±6.97a | 23.1±14.66A | 58.53±25.65c | 44.97±19.65b | 51.75±23.75B |
| BAC | 10$^6$/g | 1.31±3.01a | 2.97±7.23ab | 2.14±5.58A | 2.00±2.21ab | 4.73±11.27b | 3.36±8.20A |
| FUN | 10$^4$/g | 0.82±1.34a | 0.71±0.90a | 0.76±1.14A | 1.27±1.93a | 0.92±1.45a | 1.09±1.71A |
| ACT | 10$^5$/g | 0.49±1.25ab | 0.92±1.75b | 0.71±1.53A | 0.55±1.27ab | 0.18±0.28a | 0.37±0.94B |

Notes: Different lower-case (or upper-case) letters indicate significant difference at $P < 0.05$ (one-way ANOVA). The measured soil
properties are: WC, soil water content; SOM, soil organic matter; TN, total nitrogen; NN, nitrate nitrogen; AN, ammonium nitrogen; TP,
total phosphorus; AP, available phosphorus; TK, total potassium; AK, available potassium; MBC, microbial biomass carbon; RQ,
microbial respiratory quotient; CD, cellulose decomposition; NF, nitrogen fixation; UR, urease; ACP, acid phosphatase; CAT, catalase;
CEL, cellulose ; SI, sucrose invertase; BAC, bacteria; FUN, fungi; ACT, actinomyces.





Table 3 Results of principal component analysis and weight values of each soil
property.

| Soil properties | PC1 | PC2 | PC3 | PC4 | PC5 | PC6 | Norm | Communalities | Weight1 | Weight2 |
|---|---|---|---|---|---|---|---|---|---|---|
| WC | 0.457 | -0.562 | 0.011 | 0.383 | -0.030 | 0.228 | 1.539 | 0.725 | 0.050 | |
| AN | 0.395 | 0.482 | 0.257 | 0.462 | -0.017 | -0.229 | 1.430 | 0.721 | 0.050 | |
| NN | **0.726** | -0.269 | 0.205 | -0.170 | -0.381 | -0.074 | 1.888 | 0.821 | 0.057 | |
| AP | 0.237 | 0.160 | **0.653**[a] | -0.134 | -0.514 | 0.268 | 1.293 | 0.862 | 0.060 | 0.160 |
| AK | 0.550 | **0.631**[a] | -0.043 | -0.134 | 0.119 | -0.189 | 1.722 | 0.771 | 0.054 | 0.161 |
| SOM | **0.838** | -0.080 | 0.013 | 0.101 | 0.247 | 0.101 | 2.044 | 0.791 | 0.055 | |
| PH | -0.496 | 0.484 | 0.266 | 0.079 | 0.206 | 0.365 | 1.563 | 0.733 | 0.051 | |
| TN | **0.777** | -0.071 | 0.487 | -0.115 | 0.017 | -0.006 | 1.994 | 0.859 | 0.060 | |
| TP | 0.102 | -0.476 | 0.427 | 0.208 | 0.446 | -0.084 | 1.176 | 0.668 | 0.046 | |
| TK | -0.244 | 0.789 | -0.014 | -0.089 | 0.024 | 0.134 | 1.464 | 0.708 | 0.049 | |
| MBC | 0.051 | 0.371 | 0.291 | **0.617**[a] | -0.275 | 0.015 | 1.092 | 0.681 | 0.047 | 0.158 |
| RQ | 0.251 | 0.129 | -0.610 | 0.238 | -0.132 | 0.293 | 1.153 | 0.611 | 0.042 | |
| CD | -0.358 | -0.158 | -0.028 | **0.501**[a] | 0.138 | -0.052 | 1.089 | 0.426 | 0.030 | 0.083 |
| UR | 0.650 | 0.092 | 0.211 | -0.055 | **0.502**[a] | -0.276 | 1.723 | 0.806 | 0.056 | 0.120 |
| ACP | **0.830**[a] | 0.275 | -0.269 | -0.078 | 0.040 | -0.008 | 2.084 | 0.845 | 0.059 | 0.162 |
| CAT | 0.619 | 0.221 | -0.349 | 0.097 | 0.161 | 0.346 | 1.664 | 0.708 | 0.049 | |
| SI | 0.733 | 0.083 | 0.004 | -0.135 | -0.146 | 0.069 | 1.783 | 0.589 | 0.041 | |
| CEL | **0.793** | -0.018 | -0.271 | 0.173 | -0.053 | 0.069 | 1.955 | 0.740 | 0.051 | |
| BAC | -0.006 | 0.250 | 0.168 | -0.284 | 0.360 | 0.298 | 0.786 | 0.390 | 0.027 | |
| FUN | 0.019 | 0.275 | -0.255 | 0.007 | -0.178 | **-0.629**[a] | 0.920 | 0.568 | 0.039 | 0.156 |
| ACT | -0.208 | 0.490 | 0.206 | 0.189 | 0.059 | -0.053 | 1.035 | 0.368 | 0.026 | |
| Eigenvalue | 5.771 | 2.838 | 1.931 | 1.377 | 1.308 | 1.168 | | | | |
| % of Variance | 27.481 | 13.514 | 9.196 | 6.556 | 6.231 | 5.561 | | | | |
| Cumulative % | 27.481 | 40.995 | 50.191 | 56.747 | 62.978 | 68.539 | | | | |

Notes: Bold font values are considered highly weighted. [a] Values are the most important properties for the results of SQI-M.
Weight 1 refers to total data set; Weight 2 refers to the important properties data set. The measured soil properties are: WC, soil
water content; SOM, soil organic matter; TN, total nitrogen; NN, nitrate nitrogen; AN, ammonium nitrogen; TP, total phosphorus;
AP, available phosphorus; TK, total potassium; AK, available potassium; MBC, microbial biomass carbon; RQ, microbial
respiratory quotient; CD, cellulose decomposition; NF, nitrogen fixation; UR, urease; ACP, acid phosphatase; CAT, catalase; CEL,
cellulose ; SI, sucrose invertase; BAC, bacteria; FUN, fungi; ACT, actinomyces.



Table 4 Correlation coefficients among the soil properties.

| | WC | AN | NN | AP | AK | SOM | pH | TN | TP | TK | MBC | RQ | CD | UR | ACP | CAT | SI | CEL | BAC | FUN | ACT |
|---|---|---|---|---|---|---|---|---|---|---|---|---|---|---|---|---|---|---|---|---|---|
| WC | 1 | 0.00 | 0.37** | 0.02 | -0.17** | 0.51** | -0.35** | 0.37** | 0.26** | -0.51** | 0.07 | 0.12 | 0.06 | 0.15* | 0.22** | 0.22** | 0.25** | 0.37** | -0.15* | -0.19** | -0.30** |
| AN | | 1 | 0.16** | 0.19** | 0.45** | 0.28** | 0.03 | 0.30** | 0.00 | 0.13* | 0.49** | 0.05 | -0.07 | 0.35** | 0.34** | 0.22** | 0.18** | 0.32** | 0.02 | 0.12* | 0.19** |
| NN | | | 1 | 0.44** | 0.20** | 0.50** | -0.56** | 0.66** | 0.12 | -0.37** | -0.04 | 0.03 | -0.27** | 0.30** | 0.52** | 0.22** | 0.52** | 0.52** | -0.09 | -0.01 | -0.22** |
| AP | | | | 1 | 0.13* | 0.08 | 0.14* | 0.49** | 0.01 | 0.10 | 0.23** | -0.14* | -0.19** | 0.00 | 0.02 | -0.01 | 0.27** | 0.08 | 0.07 | -0.10 | 0.08 |
| AK | | | | | 1 | 0.39** | -0.07 | 0.38** | -0.23** | 0.38** | 0.12 | 0.15* | -0.27** | 0.50** | 0.67** | 0.37** | 0.41** | 0.34** | 0.07 | 0.16** | 0.12 |
| SOM | | | | | | 1 | -0.26** | 0.67** | 0.21** | -0.26** | -0.01 | 0.17** | -0.21** | 0.56** | 0.68** | 0.53** | 0.53** | 0.66** | 0.02 | -0.04 | -0.17** |
| pH | | | | | | | 1 | -0.26** | -0.07 | 0.50** | 0.14* | -0.16* | 0.11 | -0.26** | -0.34** | -0.06 | -0.34** | -0.47** | 0.19** | -0.03 | 0.36** |
| TN | | | | | | | | 1 | 0.29** | -0.20** | 0.06 | -0.05 | -0.30** | 0.63** | 0.48** | 0.24** | 0.59** | 0.43** | 0.05 | -0.08 | -0.14* |
| TP | | | | | | | | | 1 | -0.38** | -0.07 | -0.16* | 0.08 | 0.28** | -0.19** | -0.07 | -0.08 | 0.05 | -0.04 | -0.17** | -0.42 |
| TK | | | | | | | | | | 1 | 0.17** | 0.08 | 0.01 | -0.08 | 0.00 | -0.00 | -0.09 | -0.19** | 0.14* | 0.08 | 0.32** |
| MBC | | | | | | | | | | | 1 | 0.04 | 0.04 | 0.02 | 0.04 | 0.04 | -0.01 | 0.01 | 0.02 | 0.06 | 0.15* |
| RQ | | | | | | | | | | | | 1 | -0.05 | -0.06 | 0.31** | 0.36** | 0.17** | 0.40** | -0.04 | 0.09 | -0.06 |
| CD | | | | | | | | | | | | | 1 | -0.17** | -0.33** | -0.25** | -0.20** | -0.18** | -0.08 | -0.05 | 0.08 |
| UR | | | | | | | | | | | | | | 1 | 0.49** | 0.32** | 0.41** | 0.37** | 0.12 | -0.00 | -0.09 |
| ACP | | | | | | | | | | | | | | | 1 | 0.64** | 0.59** | 0.66** | 0.07 | 0.13* | -0.09 |
| CAT | | | | | | | | | | | | | | | | 1 | 0.45** | 0.61** | 0.03 | -0.00 | -0.08 |
| SI | | | | | | | | | | | | | | | | | 1 | 0.52** | -0.01 | 0.02 | -0.04 |
| CEL | | | | | | | | | | | | | | | | | | 1 | -0.07 | 0.06 | -0.13* |
| BAC | | | | | | | | | | | | | | | | | | | 1 | -0.04 | 0.06 |
| FUN | | | | | | | | | | | | | | | | | | | | 1 | 0.09 |
| ACT | | | | | | | | | | | | | | | | | | | | | 1 |

Notes: * P < 0.05, ** P < 0.01. The measured soil properties are: WC, soil water content; SOM, soil organic matter; TN, total
nitrogen; NN, nitrate nitrogen; AN, ammonium nitrogen; TP, total phosphorus; AP, available phosphorus; TK, total potassium;
AK, available potassium; MBC, microbial biomass carbon; RQ, microbial respiratory quotient; CD, cellulose decomposition; NF,
nitrogen fixation; UR, urease; ACP, acid phosphatase; CAT, catalase; CEL, cellulose ; SI, sucrose invertase; BAC, bacteria; FUN,
fungi; ACT, actinomyces.

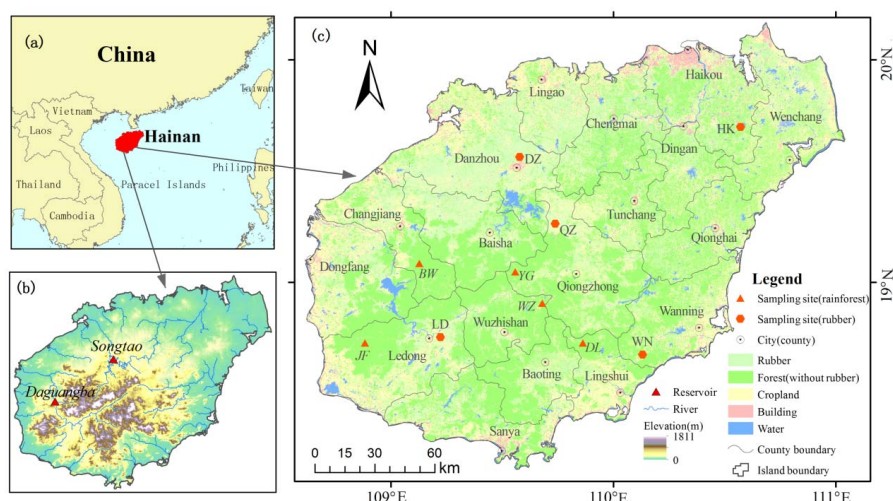


**Fig.1** Maps of the geographic position, topography, and soil sampling sites of Hainan

Island, China: (a) location of Hainan Island (red); (b) topography and drainage of

Hainan Island; (c) spatial distribution of soil sampling sites in tropical rainforests and

rubber plantations.




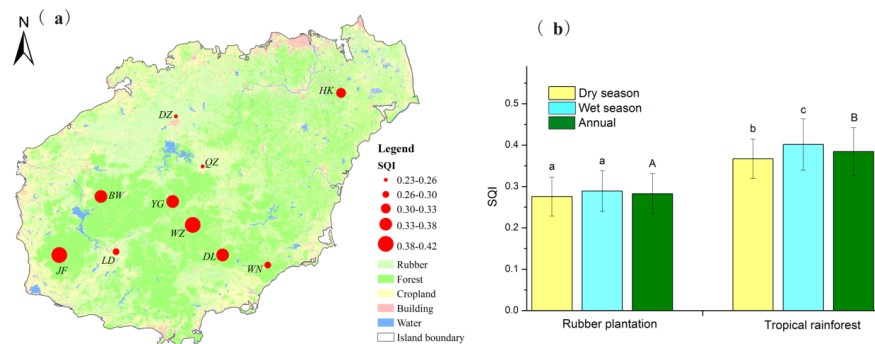


**Fig. 2** Soil quality index (SQI) values under rubber plantations and tropical rainforests
on Hainan Island: (a) spatial distribution; (b) temporal variation. Different lower-case
(or upper-case) letters indicate significant difference at $P < 0.05$ between the seasonal
(or annual) SQI values of rubber plantations and tropical rainforests.


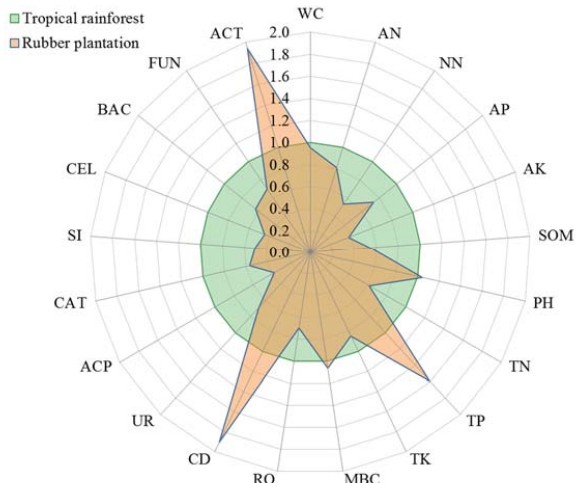


**Fig.3** Radar diagram for soil properties changing by the conversion from tropical

rainforests to rubber plantations on Hainan Island. The measured soil properties are:

WC, soil water content; SOM, soil organic matter; PH, soil pH; TN, total nitrogen;

NN, nitrate nitrogen; AN, ammonium nitrogen; TP, total phosphorus; AP, available

phosphorus; TK, total potassium; AK, available potassium; MBC, microbial biomass

carbon; RQ, microbial respiratory quotient; CD, cellulose decomposition; NF,

nitrogen fixation; UR, urease; ACP, acid phosphatase; CAT, catalase; CEL, cellulose ;

SI, sucrose invertase; BAC, bacteria; FUN, fungi; ACT, actinomyces.

764



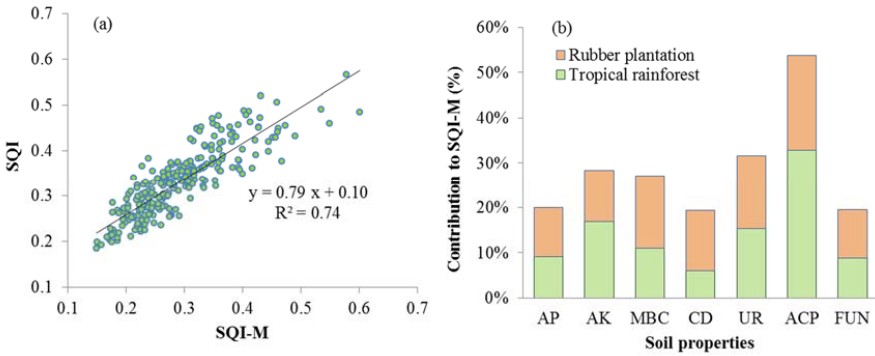

765

**Fig.4** (a) Scatter diagram and linear relationships between SQI-M and SQI values (n =

260) and (b) individual contributions of soil properties to the soil quality indicator

SQI-M based on the seven important properties (SQI is the soil quality indicator

based on the total soil properties). The measured soil properties are: AP, available

phosphorus; AK, available potassium; MBC, microbial biomass carbon; CD, cellulose

decomposition; UR, urease; ACP, acid phosphatase; FUN, fungi.

772





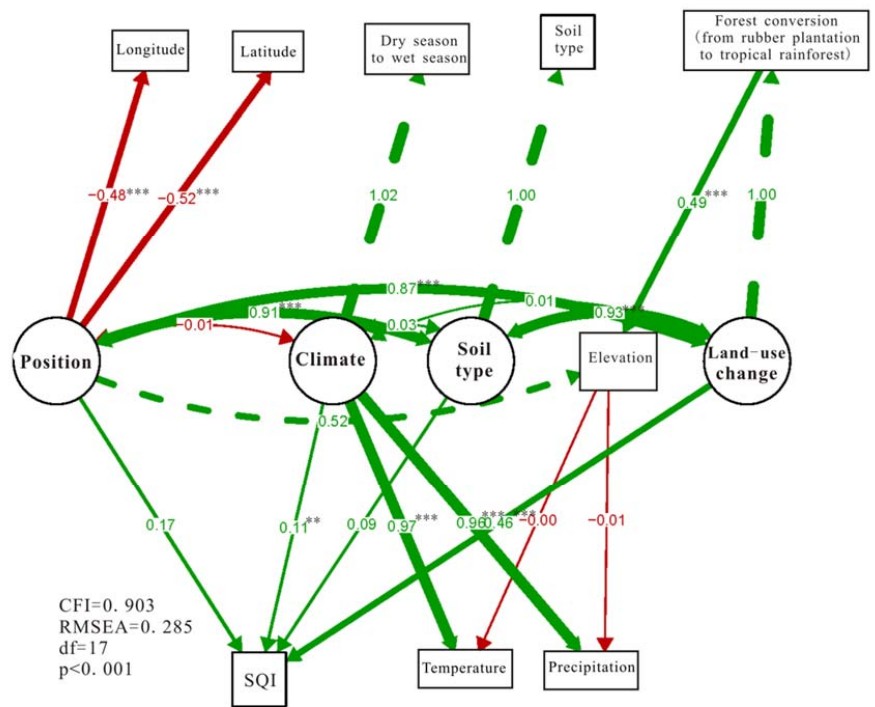

773

**Fig.5** Structural equation model (SEM) analysis of the effects of land-use changes,

soil types, climatic variables, and geographic position on the soil quality index (SQI).

Red arrows indicate negative effects and green arrows represent positive effects.

Numbers adjacent to arrows are path coefficients (*p* values) indicating the effect size

of the relationship, and *p* values are as follows: *p* <0.05; ***p* <0.01; ****p* <0.001.

CFI: Comparative Fit Index; RMSEA: Root Mean Square Error of Approximation.