# Peer review of "Effects of tropical rainforest conversion to rubber plantation"

_Biogeosciences, 2021_

## Author Comment (AC2)

Frank Hagedorn
Associate Editor
Biogeosciences
March 30, 2022

Dear Dr. Hagedorn,
Thank you very much for your help in processing our manuscript bg-2021-303. Also we like to thank the second anonymous reviewer for his/her valuable comments. Taking into account the comments, we will improve the discussion in the following three aspects.

(i) The hypotheses of this study will be discussed in detail. For example,

"SQI values were calculated ranging from 0.229 to 0.325 for rubber plantations and from 0.358 to 0.418 for tropical rainforests on Hainan Island. The averaged SQI value of rubber plantations decreased by 26.48% compared to the rainforests on the tropical island, which indicated that the soil quality would be deteriorated by the conversion of tropical rainforest to rubber plantations".

"The path coefficients between SQI and land-use change, soil types, climatic variables, and geographical position in the structural equation model are 0.46, 0.09, 0.11, and 0.17, respectively (Fig. 5), which indicated that the land-use change from rubber plantations to tropical rainforests played the most significant positive role in the spatial variation of SQI". Besides, "The path coefficients among land-use change, soil types, and geographical position were higher than 0.87, indicating that the interaction of land-use change, soil types, and geographical position was strong. While the path coefficients between the three environmental factors and climatic variables were less than 0.03, indicating their relationships were relative weak". "Overall, the spatial variation of soil quality, which was affected by the interaction of land-use change with soil types, geographical position, and climatic variables."

(ii) The soil pH, total phosphorus, cellulose decomposition, and actinomyces in rubber plantations increased by comparing with tropical rainforests on Hainan Island. And the increases were good for soil quality. These will also be discussed in detail and clearly. For example,

"The soil pH change of tropical forests to rubber plantations was consistent with the previous studies in Sumatra, Indonesia (Allen et al., 2015; 2016), which is most likely due to ashes from biomass burning of which effects on increasing soil pH can last for many years (van Straaten et al., 2015). Although there were differences in soil pH of the tropical rainforests and rubber plantations, the range of soil pH we observed was within the Al-buffering range (through Al solubility at pH 3-5; Van Breemen et al., 1983), signifying the generally low acid buffering capacity in the soils (i.e., laterite, lateritic red soil, red soil, and yellow soil) on Hainan Island".

"The soil total phosphorus in rubber plantations was greater than the tropical rainforest on Hainan Island. The greater soil total phosphorus concentrations on rubber plantations could be caused by fertilization (Yang et al., 2010; Wang et al., 2017)".

"Soil cellulose decomposition in rubber plantations increased by comparing with tropical rainforests, which can be attributed to the specific humus horizons affected by land-use type interactive with climatic and edaphic conditions (Drewnik, 2006). Besides, previous study showed that the relationship between a slower rate of cellulose decomposition and increasing altitude is not absolute (Drewnik, 2006)".

"Soil actinomyces quantities in rubber plantations was significantly higher than the tropical rainforests on Hainan Island. It could be because of the soil nutrient and aeration environment were more conducive to actinomycetes growth in rubber plantations, and they also showed a positive correlation (Pang et al., 2009; Fan et al., 2021). In addition, inhibition of other species in the soil on actinomycetes may also relative weak, leading to a rise in the number of actinomyces in rubber plantations (Gong et al., 2018)".

(iii) The seasonal difference of soil quality and soil properties will also be discussed more deeply. For example,

"Seasonal changes also played a role in soil quality. According to the SQI values, tropical rainforests in the wet season were significantly higher than those in the dry season. However, there were no significant differences in SQI values for rubber plantations, which can be attributed to the fertilization in dry seasons. Soil NN was the only soil property which showed the similar seasonal changes with the SQI for both tropical rainforests and rubber plantations on Hainan Island. That is, the soil NN in wet season was significantly higher than those in dry season for tropical rainforests, while there was little seasonal difference for rubber plantations due to the fertilization. Although the effect of seasonal change on the SQI values under rubber plantations was relatively small, it controlled some important soil chemical and biological properties (e.g., TN, TP, AP, URE, and SI) as well as the bacterial communities in soils of rubber plantations in tropical region of Hainan (Lan et al., 2018; 2020)".

Main references:

1. Allen, K., Corre, M.D., Kurniawan, S., Utami, S.R., Veldkamp, E., 2016. Spatial variability surpasses land-use change effects on soil biochemical properties of converted lowland landscapes in Sumatra, Indonesia. Geoderma 284, 42-50.
2. Allen, K., Corre, M.D., Tjoa, A., Veldkamp, E., 2015. Soil nitrogen-cycling responses to conversion of lowland forests to oil palm and rubber plantations in Sumatra, Indonesia. Plos One 10(7), e0133325.
3. Drewnik, M., 2006. The effect of environmental conditions on the decomposition rate of cellulose in mountain soils, Geoderma 132, 116-130.
4. Fan, L., Tarin, M. W. K., Zhang, Y., Han, Y., Rong, J., Cai, X., Chen, L., Shi, C., Zheng, Y., 2021. Patterns of soil microorganisms and enzymatic activities of various forest types in coastal sandy land, Glob. Ecol. Conserv. 28, e01625.

5.  Gong, B., Chen, S., Lan, W.W., Huang, Y.M., Zhu, X.C., 2018. Antibacterial and antitumor potential of actinomycetes isolated from mangrove soil in the Maowei Sea of the southern coast of China. Iran. J. Pharm. Res. 17, 1339-1346.
6.  Pang, X., Ning, W., Qing, L., Bao, W., 2009. The relation among soil microorganism, enzyme activity and soil nutrients under subalpine coniferous forest in Western Sichuan, Acta Ecologica Sinica 29(5), 286-292.
7.  Van Breemen, N., Mulder, J., Driscoll, C.T., 1983. Acidification and alkalinization of soils. Plant Soil 75, 283-308.
8.  van Straaten, O., Corre, M.D., Wolf, K., Tchienkoua, M., Cuellar, E., Matthews, R.B., Veldkamp, E., 2015. Conversion of lowland tropical forests to tree cash crop plantations loses up to one-half of stored soil organic carbon. Proc. Natl. Acad. Sci. 112, 9956-9960.
9.  Wang, J., Ren, C., Cheng, H., Zou, Y., Bughio, M.A., Li, Q., 2017. Conversion of rainforest into agroforestry and monoculture plantation in China: consequences for soil phosphorus forms and microbial community. Sci. Total Environ. 595, 769-778.
10. Yang, K., Zhu, J.J., Yan, Q.L., Sun, O.J., 2010. Changes in soil P chemistry as affected by conversion of natural secondary forests to larch plantations. For. Ecol. Manag. 260, 422-428.

Sincerely,

Rui Sun